# Conserved phosphorylation hotspots in eukaryotic protein domain families

Marta J. Strumillo [1], Michaela Oplová[2,3], Cristina Viéitez[1,4], David Ochoa [1], Mohammed Shahraz[4], Bede P. Busby [1,4], Richelle Sopko[5], Romain A. Studer[1,8], Norbert Perrimon [5,6,7], Vikram G. Panse[2] & Pedro Beltrao [1]

Protein phosphorylation is the best characterized post-translational modification that regulates almost all cellular processes through diverse mechanisms such as changing protein conformations, interactions, and localization. While the inventory for phosphorylation sites across different species has rapidly expanded, their functional role remains poorly investigated. Here, we combine 537,321 phosphosites from 40 eukaryotic species to identify highly conserved phosphorylation hotspot regions within domain families. Mapping these regions onto structural data reveals that they are often found at interfaces, near catalytic residues and tend to harbor functionally important phosphosites. Notably, functional studies of a phospho-deficient mutant in the C-terminal hotspot region within the ribosomal S11 domain in the yeast ribosomal protein uS11 shows impaired growth and defective cytoplasmic 20S pre-rRNA processing at 16 °C and 20 °C. Altogether, our study identifies phosphorylation hotspots for 162 protein domains suggestive of an ancient role for the control of diverse eukaryotic domain families.

[1] European Molecular Biology Laboratory, European Bioinformatics Institute, Wellcome Genome Campus, Hinxton, Cambridge CB10 1SD, UK. [2] Institute of Medical Microbiology, University of Zurich, CH-8006 Zurich, Switzerland. [3] Institute of Biochemistry, ETH Zurich, CH-8093 Zurich, Switzerland. [4] European Molecular Biology Laboratory, Genome Biology Unit, 69117 Heidelberg, Germany. [5] Department of Genetics, Harvard Medical School, 77 Avenue Louis Pasteur, Boston, MA 02115, USA. [6] Drosophila RNAi Screening Center, Harvard Medical School, 77 Avenue Louis Pasteur, Boston, MA 02115, USA. [7] Howard Hughes Medical Institute, 77 Avenue Louis Pasteur, Boston, MA 02115, USA. [8] Present address: BenevolentAI, London NW1 1LW, UK. Correspondence and requests for materials should be addressed to P.B. (email: pbeltrao@ebi.ac.uk)

Protein post-translational regulation is a reversible and highly effective way to regulate protein functions within seconds to minutes time scales. Over 300 different types of protein post-translational modifications (PTMs) are known, ranging from single atom modifications (oxide) to small protein modifiers (ubiquitin)[1]. Mass spectrometry and biochemical enrichment methods allow for the study of PTM regulation at very large scale[2] and such approaches have been extensively applied to study protein phosphorylation[3]. On the most comprehensive single study to date on the order of 75% of the detected proteome was found to be phosphorylated[4] and approximately 160,000 non-redundant human phosphosites are listed in public repositories[5]. Although the regulation of protein functions by phosphorylation has been under study for over 60 years[6] the recent wealth of knowledge regarding protein phosphorylation generated by mass-spectrometry remains mostly uncharacterized. Only around 5% of human phosphosites have an annotated regulatory role[5]. Evolutionary studies have suggested that some degree of protein phosphorylation, and other PTM sites, may have little to no biological function[7,8]. Given that some phosphosites may have no relevance to fitness, devising ways to rank sites according to functional importance is a crucial research question. Functionally important phosphosites have been shown to be more likely conserved across species and across protein domains of the same type[8,9]. For protein domains, conserved phosphorylation within a specific region—termed phosphorylation hotspot—tends to identify regions with regulatory potential[8]. Since each domain family is represented by multiple copies within each species, these domain-centric analyses have the advantage of increased statistical power when compared to the study of conservation of orthologous genes. Domain-centric analyses have also been used with success to study recurrence of mutations in cancer samples[10].

We have previously studied domain phosphorylation hotspots across 10 example domain types[8]. To perform a systematic analysis we compile here an expanded set of over 500,000 phosphosites across 40 eukaryotic species. These phosphosites are mapped to protein domain regions allowing us to identify phosphorylation hotspots within 162 domain families. We show that the identified regions are functionally relevant as they are enriched in previously known regulatory phosphosites, interaction residues, and positions that are close to catalytic residues. We identify a putative regulatory region in the ribosomal S11 domain and generate a phosphorylation-deficient mutant in two *Saccharomyces cerevisiae* phosphosites found within this region of the yeast ribosomal protein Rps14A. We show that the Rps14a-T119A mutant exhibits impaired growth at 16 and 20 °C, and is defective in cytoplasmic 20S pre-rRNA processing, uncovering a critical role for phosphorylation of this region during eukaryotic ribosome assembly.

## Results

**Eukaryotic phosphorylation hotspot domain regions**. In order to study the conservation of protein phosphorylation within protein domain families, we collected protein phosphosite data from publicly available sources for a total of 40 eukaryotic species, including 11 animals, 19 fungi, 7 plants, and 3 apicomplexa species (Fig. 1a and Methods). A total of 537,321 phosphosites were compiled and mapped to reference proteomes and protein domain regions were identified using the Pfam domain[11] models across all species (Methods) and phosphosites were matched to these regions. Of all phosphosites, 83,359 phosphosites occur within Pfam domain regions (Fig. 1a). As most phosphosites tend to occur in disordered

regions[12] it is not unexpected that the majority of sites are not found within protein domains. The ranked list of most commonly modified domains is shown in Supplementary Table 1. In line with previous findings, the most commonly regulated domains included many involved in cell signaling (e.g., protein kinase, Ras), chaperone function (e.g., HSP70, TCP, HSP90), and cytoskeleton (e.g., Actin, Myosin).

In order to statistically identify domain regions that are regulated by phosphorylation above random expectation, we selected 344 domain families that are represented by at least 10 different instances and contained a total of 50 or more phosphosites. For these domain families, the protein sequences containing phosphosites were aligned and an enrichment score was calculated using a rolling window approach—with a fixed length of 5 positions—to identify regions with an above average degree of phosphorylation as illustrated in Fig. 1b. The random expectation was calculated by permutation testing where phosphosites were randomly re-assigned within each protein sequence to equivalent phospho-acceptor residues (Methods). A rolling window approach was used to take into account alignment uncertainty and errors in assignment of the phosphosite position within the phosphopeptide as identified in the mass spectrometry studies. For each position within the domain alignments a $p$-value was calculated and after Bonferroni multiple testing correction a corrected cut-off $p$-value <0.01 (uncorrected $p$-value $6.70 \times 10^{-8}$) was used globally to identify domain regulatory hotspots. A cut-off of an average of 2 phosphosites per position was also used to avoid significant positions with a low effect size difference. Contiguous positions were merged to identify domain regions of interest that were defined as phosphorylation hotspot regions (Methods). Using this procedure a total of 1999 positions corresponding to 241 domain regions were identified as hotspots within 162 Pfam domain families (listed in Supplementary Data 1).

**Validation of phosphorylation hotspot regions**. Under the assumption that strong conservation of protein phosphorylation within a region of a domain is predictive of functional relevance we would expect to find an enrichment of phosphosites with known functions at predicted phosphorylation hotspot regions. We therefore tested if the hotspot predictions could discriminate between human phosphosites with known function from other human phosphosites with unknown function. For each human phosphosite within the protein domains analyzed we assigned the hotspot $p$-value. We considered only Pfam domains having more than 1 known human regulatory phosphosites and we analyzed separately serine/threonine (S/T) sites from tyrosines (Y). We were able to analyze a set of 983 S/T and 317 Y phosphosites with known regulatory functions in human, as defined in PhosphoSitePlus, out of a total of 8270 S/T and 1395 Y human phosphorylated positions within the same domains.

The capacity to discriminate the human phosphosites with known regulatory roles from other human phosphosites was tested using the receiver operating characteristic (ROC) curve (Fig. 1c) and summarized as the area under the ROC (AUC) curve. The regulatory hotspot $p$-value was a strong predictor of know regulatory phosphosites (AUC = 0.78 for S/T and 0.62 for Y). In line with this, the defined hotspot regions show significant enrichment over random for human phosphosites of known function (Fig. 1d). Overall, these results show that the regulatory hotspots identified here are enriched in previously known regulatory phosphosites. For commonly used model species we listed in Supplementary Data 1 the currently known phosphosites that are found within these regions as these are more likely to have important functional roles.

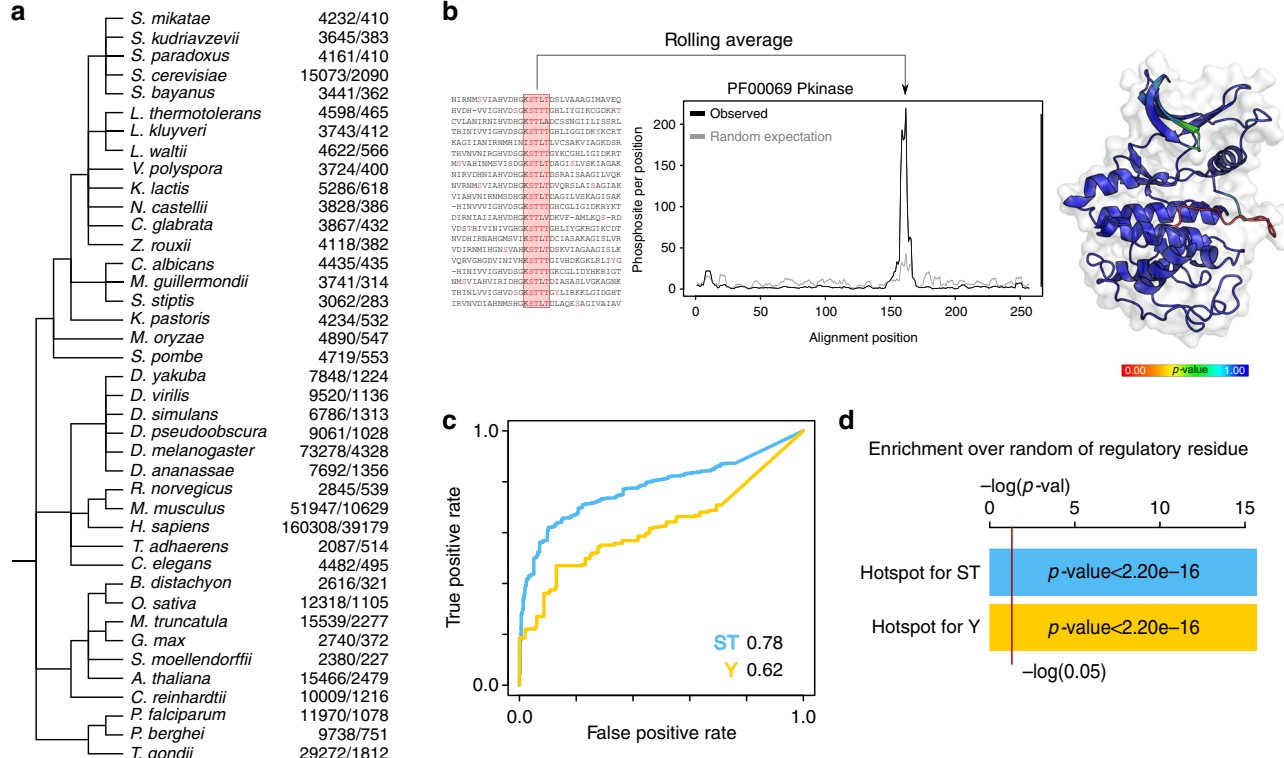

**Fig. 1** Prediction of phosphorylation hotspots regions for eukaryotic domain families. **a** Phylogenetic tree of the species from which phosphorylation data has been obtained. The numbers in the left column correspond to the phosphosites per species obtained and the right column the phosphosites found within Pfam domains. **b** Hotspot regions are defined as those having higher than randomly expected number of phosphorylation. A rolling window is used to count the observed average number of phosphosites in the alignment (black line) and a background expectation is calculated from random sampling (gray line and gray band for standard deviation). A p-value is calculated for the enrichment of phosphorylation at each position and projected onto structural models. **c** The capacity to discriminate between phosphosites of known function from other phosphosites was tested using a ROC curve. We compared the discrimination power of the hotspot p-value (blue line for ST and yellow line for Y). **d** Enrichment over random of human phosphosites with known functions for residues predicted as a hotspot region when compared with the rest of the domain (blue for ST and yellow for Y; p-values for Fisher's exact test)

**Mapping of regulatory hotspots to structural models**. The phosphorylation hotspots are of functional and structural interest as these are very likely to have regulatory potential that should often be a general property of the domain family. To study these regions in the context of protein structures we have collected structures available for Pfam domains in PDB[13]. For each domain family we discarded structures with short sequences, performed structural clustering, and selected a representative from the most populated structural cluster (Methods). The protein sequences from the selected structural models were added to the alignments and a total of 116 hotspots regions were mapped to a 3D model for 85 domains. We provide the structural representation of these hotspot regions and enrichment plots in Supplementary Data 2. To gain a better understanding of how these regions may control domain functions we studied in more detail some regions that overlap with human phosphosites of known function (Fig. 2).

The protein kinase activation loop is the prototypical example of a regulatory hotspot (Fig. 2, top). Over 50% of phosphosites and 74% (128 out 174) of known human regulatory sites (red dots) found within protein kinases are in this loop that follows the β9 sheet near the active site. Phosphorylation of this loop is typically required to achieve full activation of kinases by positioning the activation segment in order to allow for substrate recognition[14]. Another well characterized example is the regulation of the pyruvate dehydrogenase complex (PDC) which is primarily composed of multiple copies of pyruvate dehydrogenase (E1), dihydrolipoamide acetyltransferase (E2), and

dihydrolipamide dehydrogenase (E3). PDC activity is regulated by phosphorylation of E1 in positions that overlap with our identified phosphorylation hotspot (Fig. 2). The phospho-regulation of this domain is well characterized with 3 described regulatory phosphosites[15,16]. Two of these positions fall within what is termed the phosphorylation loop A (Ph-loop A) region, which overlaps directly with the hotspot region. Phosphorylation of this loop region is known to induce a conformational change in the loop that causes enzyme inhibition[17]. We expect that the identified hotspots from other domain families will be of regulatory importance in analogy to the activation loop of kinases and the phosphorylation loops of pyruvate dehydrogenase.

A clear phosphorylation hotspot was found for the Ras domain family (Fig. 2). This small GTPase superfamily is known to change in conformation depending on the GTP versus GDP bound state with two loop regions—called switch 1 and switch 2—being particularly sensitive to the nucleotide binding. The major Ras phosphorylation hotspot occurs just after the switch 2 region at the start of α2. This region is also known to often form contacts with effector molecules[18] implicating the phosphorylation of this region in the regulation of Ras-effector interactions. Supporting this hypothesis, the phosphorylation of human KRas in this region at Y64 regulates the interaction between KRas and AGO2[19]. Similarly, phosphorylation of the same region in Rab7 (S72) and in RAC1 (Y64) is associated with changes in interaction partners[20,21]. We suggest that this is a commonly used regulatory feature of Ras domains.

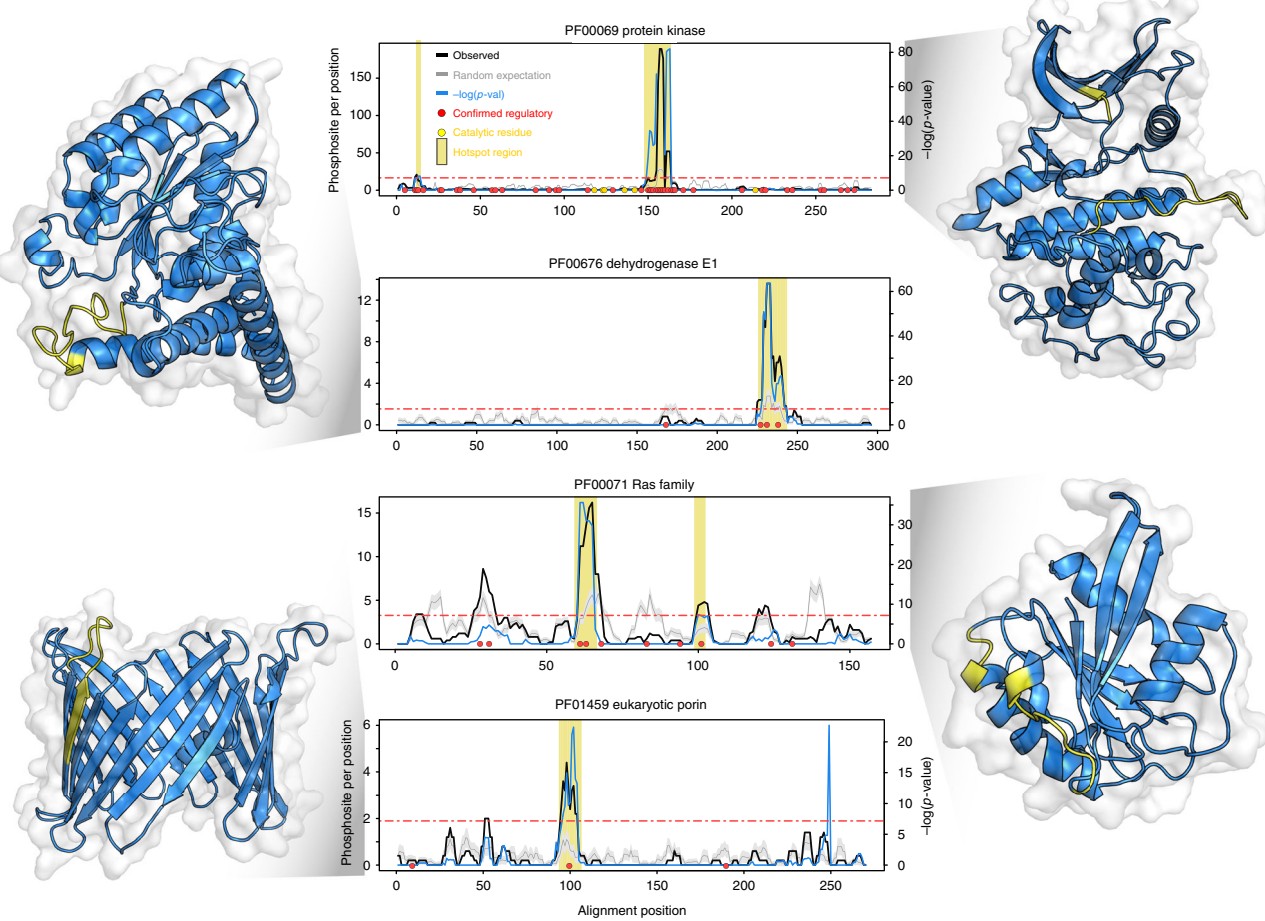

**Fig. 2** Phosphorylation hotspots overlapping with human phosphosites of known function. For 4 protein domain families we show the enrichment over random of protein phosphorylation along the domain sequence. The average number of phosphosites observed per rolling window is plotted in a solid black line (observed). The background level of expected phosphorylation calculated from random sampling is shown in gray line, with standard deviations as gray band. The blue line represents the negative logarithm of $p$-value at each position (right $y$ axis). A horizontal red line indicates a cut-off of the Bonferroni corrected $p$-value of 0.01. Positions with a $-\log(p\text{-value})$ above this cut-off and average phosphosites per window higher than 2 are considered putative regulatory regions and highlighted under a vertical yellow bar. Red circles indicate human phosphosite positions with known regulatory function. In the structural representations the predicted hotspot regions are highlighted in yellow

A different mode of phospho-regulation has been observed in the voltage-dependent anion channel (VDAC). This 19 beta sheet beta barrel domain is a class of porin ion channel spanning the outer mitochondrial membrane. The major phosphorylation hotspot in the VDAC domain is predicted for a loop region between β6 and β7 (Fig. 2). The regulation of the human VDAC1 at position S104 within this region controls VDAC1 protein levels by inhibiting proteasome-mediated degradation[22]. The orientation of VDAC domains in the membrane is contentious with conflicting reports suggesting that the C-terminal may point towards the cytoplasm[23], the mitochondrial inter-membrane space (IMS) or that is may occur in both orientations[24]. The phosphorylation hotspot between β6 and β7 is on the opposite side of C-terminal region, suggesting that this loop most likely or most often will face the cytoplasm placing the C-terminal towards the IMS.

These examples further illustrate how our analysis recovers well known regulatory regions as well as shows some ways in which domain function is regulated by protein phosphorylation (e.g., changing activity, conformation, interactions, degradation rates). We next tested if some of the regulatory mechanisms found in these examples can be generalizable to other domain families.

**Structural characterization of phosphorylation hotspots**. Regulation of interactions and catalytic activities may be general mechanisms of domain regulation. To study this across all domains we used annotations for interface residues from the 3DID database[25] and for catalytic residues based on UniProt annotations[26] (Methods). In addition, we also analyzed surface accessibility, defined as >20 relative surface accessibility (RSA), and predicted disorder (from DISOPRED[27] (Methods)). As expected from prior studies of protein phosphorylation[28] the hotspot positions are more likely to be surface exposed (Fig. 3a, Fisher's exact test, $p$-value $= 1.66 \times 10^{-8}$) and within disordered elements (Fig. 3a, Fisher's exact test, $p$-value $< 2.20 \times 10^{-16}$) when compared to other residues. We next measured distances between hotspot positions and catalytic residues. Compared to other residues within enzymes, hotspot positions are 3 times more likely to be catalytic residues (Fig. 3a, catalytic residues, $p$-value $= 0.03$); 3.4 times more likely to be within 5 amino-acid distance (Fig. 3a, Fisher's exact test, $p$-value $= 1 \times 10^{-8}$) and 5 times more likely to be within 5 Å distance (Fig. 3a, <5 Å, Fisher's exact test, $p$-value $= 1.5 \times 10^{-8}$) to catalytic residues. For enzyme domains 3.3% of hotspot residues are within 5 Å distance of catalytic residues compared with 0.97% for other residues.

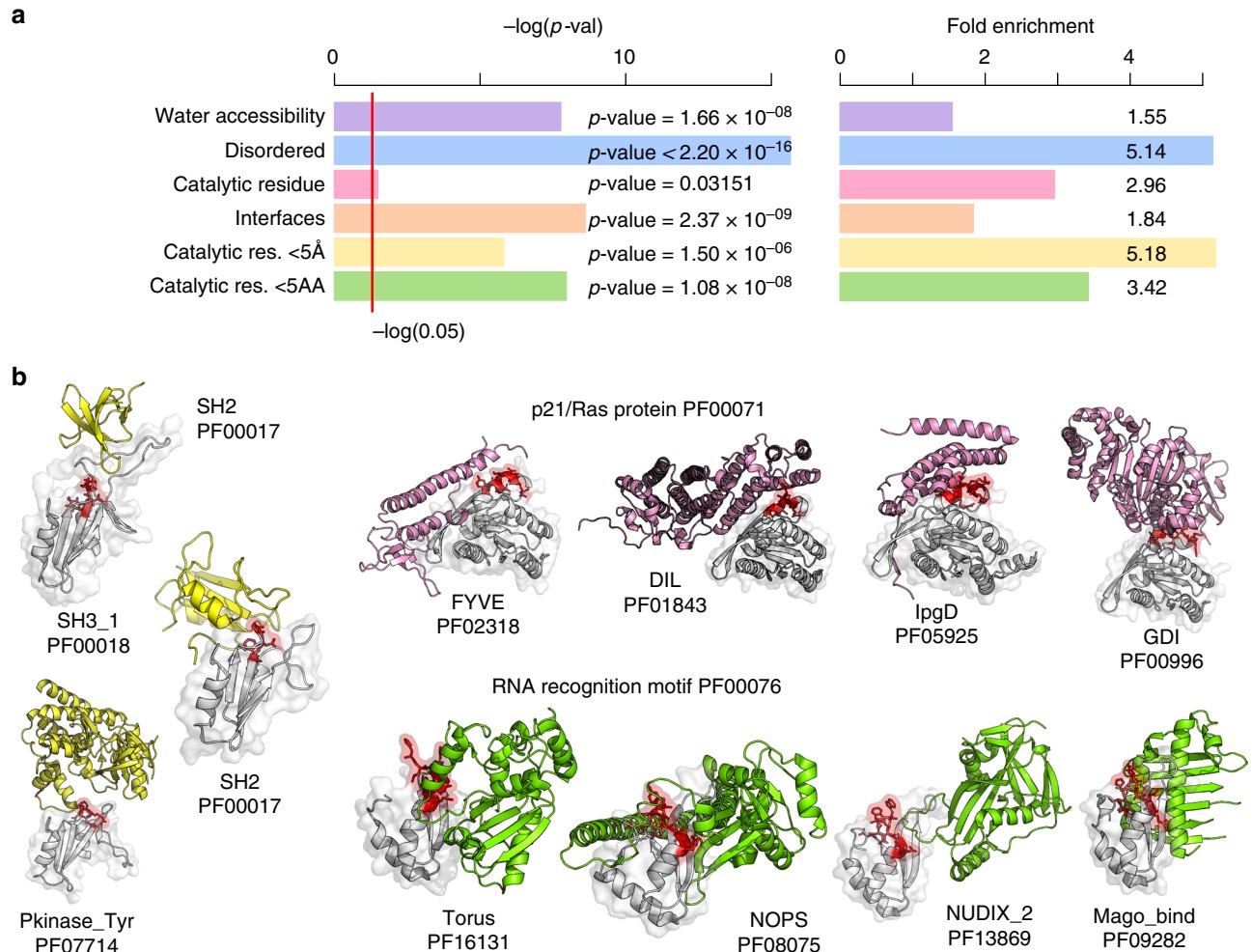

**Fig. 3** Structural features of phosphorylation hotspots. **a** Enrichment over random of structural features comparing hotspots with other residues within the same domains. The tested features include water accessibility—residues with >20% RSA; protein disorder as predicted by DISOPRED; catalytic residues; residues within 5 amino-acid distance to a catalytic residue; residues within 5 Å of a catalytic residue; residues at an interface based on 3DID. For each feature we report the −log(p-value), p-value calculated using Fisher's exact test. **b** Examples of hotspot regions at interfaces where the hotspot region (red) from a domain (gray) has been observed contacting many other types of domains (other colors) in empirical structures

For each domain position we identified interface contacts found in 3D structures with any other protein domain based on 3DID, excluding intra-domain contacts. For the interface residue enrichment test, we considered only surface accessible residues (>20% RSA) to avoid an enrichment simply due to accessibility of both interface and hotspot positions. Controlling for surface accessibility hotspots are 1.8 more likely to be interface positions (Fig. 3a, Fisher's exact test, $p$-value $< 2.4 \times 10^{-9}$). 39% of accessible hotspot residues are interface positions compared with 26% for other accessible residues. Some hotspot regions overlap with interaction regions that can make contacts with a large number of different types of domains as determined by crystal structures. For example, the hotspot region in Ras described above contacts 42 other domain types, some of which are illustrated in Fig. 3b. There are 13 domain families with hotspot regions contacting with more than 3 other domain families (Supplementary Data 1), including the SH2 domain and RNA recognition domain families shown in Fig. 3b. This suggests that protein phosphorylation of such regions may be important for switching the interaction partners of these domain families.

These results indicate that regulation of protein interactions and catalytic activities may be a recurrent feature of domain regions regulated by protein phosphorylation. Of the 116 hotspot regions mapped to a structural model, 97 overlap with interface residues and of 32 hotspot regions with putative catalytic residues, 23 are within 15 Å and 5 are within 5 Å to a catalytic residue. We annotated the list of hotspot regions (provided in Supplementary Data 1) with this information regarding interface positions and proximity to catalytic residues to facilitate future functional studies.

**Phosphorylation hotspot regions near catalytic residues**. Hotspot regions in enzyme domains are often found at or near catalytic residues and could, in these cases, play a role in regulating catalytic activities. From the 23 hotspots found within 15 Å of a catalytic residue we illustrate here 4 examples in more detail (Fig. 4). While protein phosphorylation is typically catalyzed by protein kinases we found examples of hotspot regions explained as reaction intermediates or auto-phosphorylation not catalyzed by kinase. For example, the hotspot region of alkaline phosphatase (ALP) overlaps directly a catalytic residue (Fig. 4). The hydrolysis and transphosphorylation of monoesters reaction takes place in the active pocket of the enzyme which contains a catalytic serine that forms a covalent serine-phosphate intermediate. This catalytic serine, located at the N-end of α5, is found phosphorylated in different species explaining the identified hotspot.

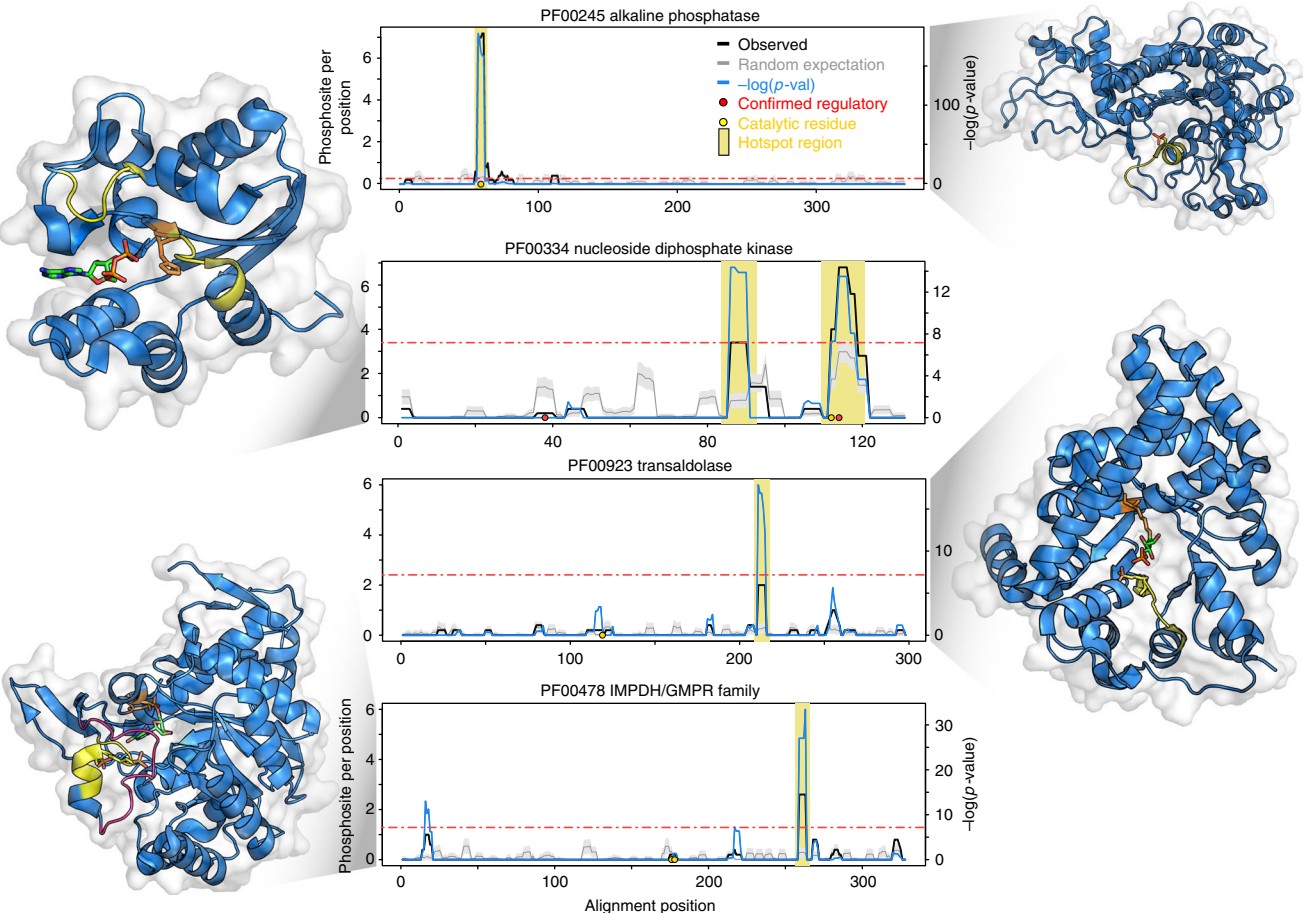

**Fig. 4** Examples of putative regulatory hotspots at or near catalytic residues. The average number of phosphosites observed per rolling window is plotted in a solid black line (observed). The background level of randomly expected phosphorylation is shown in gray line, with standard deviations as gray band. The blue line represents the negative logarithm of *p*-value at each position (right *y* axis). A horizontal red line indicates a cut-off of the Bonferroni corrected *p*-value of 0.05. Positions with a $-\log(p\text{-value})$ above this cut-off and average phosphosites per window higher than 2 are considered putative regulatory regions and highlighted under the yellow bar. Red circles indicate human phosphosite positions with known regulatory function and yellow circles represent catalytic residue positions. In the structural representations the predicted hotspot regions are highlighted in yellow. The catalytic residues have been represented in as orange sticks and in red stick representations are substrates or products

This hotspot is therefore the result of a reaction intermediate and not regulated by protein kinases. A hotspot for the phosphoglucomutase/phosphomannomutase Pfam domain (PF02878) is identical to this in that a catalytic serine is often found phosphorylated and is a reaction intermediate not catalyzed by kinases (Supplementary Fig. 1).

The nucleoside-diphosphate kinases (NDK) catalyze the exchange of terminal phosphate between different nucleoside diphosphates (NDP) and nucleoside triphosphates (NTP). A NTP serves as a donor and the reaction proceeds via a phosphohistidine intermediate in the NDK active site. The main hotspot in this domain occurs just next to this active site histidine (Fig. 4). The phospho-histidine is not detected as phosphorylated in the proteomics data, most likely due to phospho-histidine being very labile[29] and not usually searched for during the mass spectrometry data processing steps. Phosphorylation of these nearby serines has been suggested to be the result of a transfer of phosphate between the histidine and nearby serines which may be important for the enzyme activity[30,31]. In addition we found a second hotspot in the loop between α7 and α8 with an unknown function. Given that this loop partially covers the catalytic centre, the phosphorylation of this loop likely regulates substrate accessibility.

The next two domain families we studied are examples of conserved phosphorylation regions distant in sequence but close in 3D space to catalytic residues (within 15 Å). The IMP dehydrogenase (IMPDH) catalyzes the oxidation of IMP to XMP with the concomitant reduction of NAD+. In human cells Akt has been shown to interact with IMPDH and phosphorylate the protein in vitro[32] but the position or functional role of IMPDH phosphorylation has not been established. In structures of this domain a serine residue can be found in this hotspot (Fig. 5a) pointing towards the substrate binding pocket and its phosphorylation may sterically impact on substrate binding. A loop next to this hotspot changes in conformation during the catalytic cycle[33] (Fig. 5a, open to closed) so the phosphorylation of the hotspot could also have an impact on these dynamics.

Similarly to IMPDH, the hotspot region of transaldolase (TAL) is in a position that could influence the access to the catalytic centre. TAL is an enzyme of the nonoxidative part of the pentose phosphate pathway (PPP). The active site, located in the center of the barrel is formed of a lysine that holds the sugar in place and a glutamate and aspartate that act as proton donors and acceptors. There is evidence that TAL activity can be regulated by phosphorylation[34] but the position or mechanism of this regulation has not been determined. The hotspot identified for

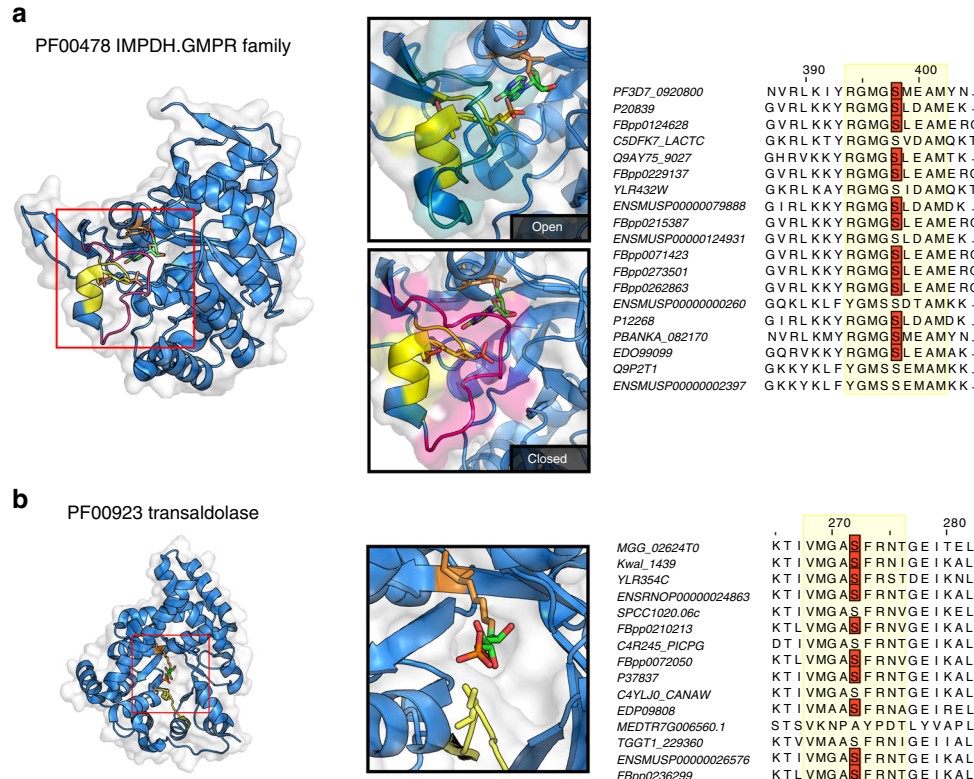

**Fig. 5** Hotspot regions near catalytic residues that are distal in protein sequence. **a** The IMPDH hotspot region is represented in yellow segment. In the insets, the loop near the hotspot region is shown changing from an open conformation (blue volumes) to closed conformation (magenta volumes). A serine residue within the hotspot region (yellow sticks) points to substrate binding pocket and is often found phosphorylated across species (see alignment). **b** The transaldolase hotspot region is shown in yellow. In the structural inset a serine within the hotspot region (yellow sticks) is found just at the entrance of the substrate cavity. The identified phosphorylation sites contributing to the identification of the hotspot region are shown in the alignments in red

TAL is very likely to alter the accessibility of the substrate to the active site. In structures of this domain a serine residue within this hotspot can be found just at the entrance to the substrate pocket (Fig. 5b) and phosphorylation of this residue may control access to the cavity.

**C-terminal hotspot region of the ribosomal S11 domain**. For further functional studies, we selected the ribosomal S11 domain family for experimental analysis of the predicted regulatory hotspot, found within the 40S ribosomal protein uS11 (yeast Rps14[35]). A phosphorylation hotspot of unknown function was identified in the C-terminal tail of this protein family (Fig. 6a). To test the functional relevance of this hotspot we selected 2 phosphosites in uS11 (yeast Rps14a) that have been identified near this region (T119 and S123), marked in Fig. 6a, b. The two sites are known to be phosphorylated in different species including human uS11 (Fig. 6c). We constructed two strains with alanine mutations at each of these positions at the genomic locus (Methods). First, we analyzed growth of these mutant strains on rich media at different temperatures and under different set of stress conditions including 6-azauracil (6AU), which depletes intracellular pools of GTP and UTP and cycloheximide (CHX), which blocks translation elongation. The T119A mutant showed an impaired growth at 16 and 20 °C on rich media (Fig. 6d, Supplementary Fig. 2a). In contrast, the S123A mutant showed only a mild growth defect at 39 °C, and on CHX, but no growth defects were observed on 6AU containing media (Supplementary Fig. 2a). Interestingly, *RPS14A* has a paralog—*RPS14B* that was not deleted or mutated for these studies, meaning that Rps14a-T119A mutant might act in a dominant negative manner.

20S pre-rRNA containing 40S pre-ribosomes assembled in the nucleolus/nucleus are exported rapidly to the cytoplasm where they undergo final maturation before achieving translation competence. One of these final steps include processing of 20S pre-rRNA into mature 18S rRNA catalyzed by the endonuclease Nob1 within an 80S-like particle formed between a mature 60S subunit and the 40S pre-ribosome[36,37]. Previous report showed that ATPase Fap7 is activated by the C-terminal tail of uS11 and this activation is critical to deposit uS11 with its neighbor eS26 on the assembling pre-ribosome[38]. Failure to deposit the ribosomal proteins uS11 and eS26 on the 40S pre-ribosome in the nucleus did not inhibit their nuclear export to the cytoplasm, but impaired cytoplasmic processing of 20S pre-rRNA into mature 18S rRNA[38,39]. Notably, point mutants within uS11 C-terminal tail also impaired 20S pre-rRNA processing in the cytoplasm[40]. We therefore investigated whether the C-terminal tail T119A and S123A mutants of uS11 were impaired in cytoplasmic processing of 20S pre-rRNA by FISH using a Cy3 labeled probe that hybridizes with ITS1 of 20S pre-RNA[35] and northern blotting. In WT cells, the Cy3-ITS1 signal is restricted to the nucleolus, since ITS1 cleaved from 20S pre-RNA in the cytoplasm is rapidly degraded by the exonuclease Xrn1 (Fig. 6e). In contrast, like in Fap7-depleted cells[38], the T119A mutant showed a strong Cy3-ITS1 signal in the cytoplasm at 20 °C (Fig. 6e). Northern analyses revealed increased 20S pre-RNA levels supporting the notion that the T119A mutation within uS11 impairs cytoplasmic processing of 20S pre-rRNA (Supplementary Fig. 2b). As expected, and like in Fap7-depleted cells[38], uS5-GFP reporter did not accumulate in the nucleus indicating that mutant strain was not defective in nuclear export of 40S pre-ribosome but impaired in final pre-

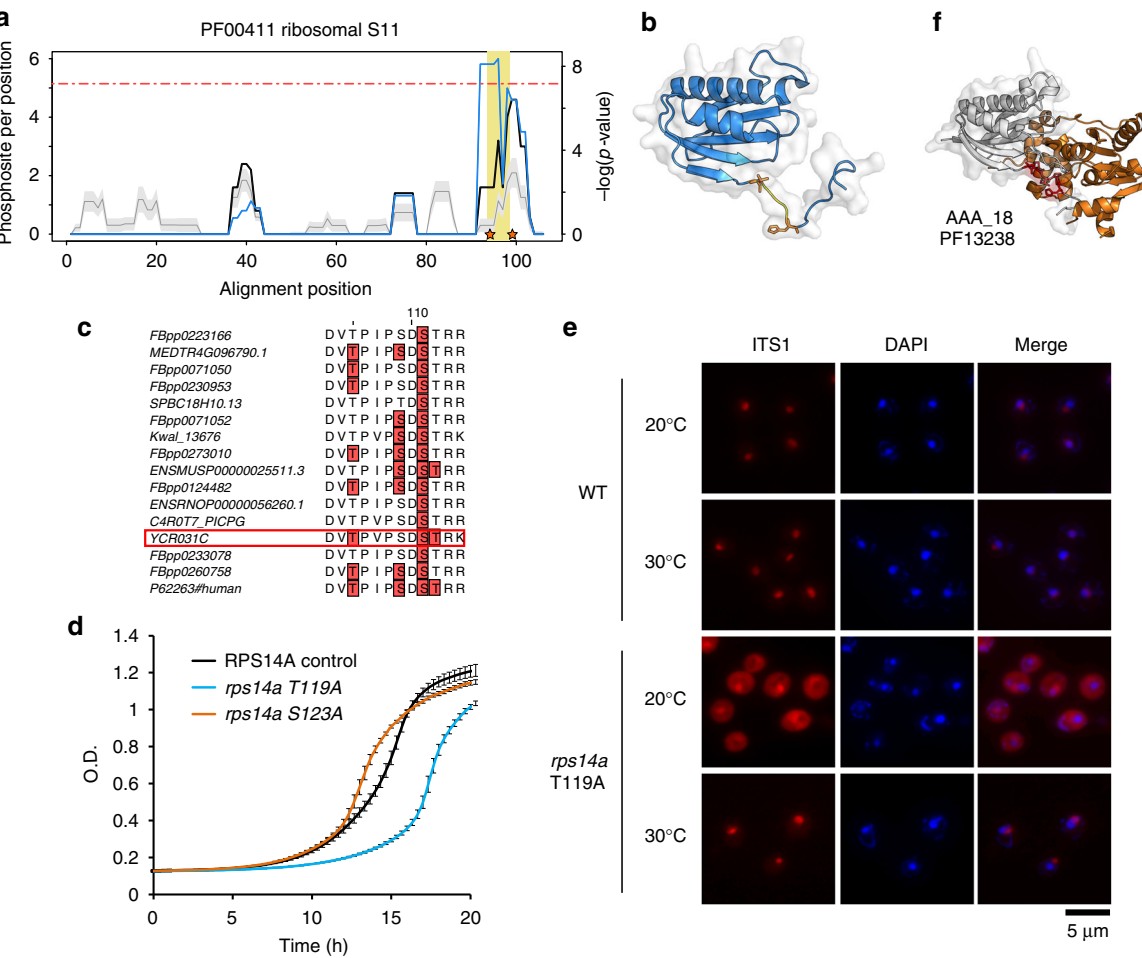

**Fig. 6** Rps14a T119A mutant shows growth and 20S processing defects in cold shock. **a** The phosphorylation enrichment over random for the ribosomal S11 domain (PFAM:PF00411) is plotted in a solid black line. The background expectation is shown in gray line, with standard deviations as gray band. The blue line represents the negative logarithm of *p*-value (*Y* axis on the right side). A horizontal red line indicates a cut-off equivalent to a Bonferroni corrected *p*-value of 0.05. Mutated residues in Rps14a (T119 and S123) are indicated by orange stars in the plot and shown in **b** as orange stick representations (PDB:5wnt_K). **c** Conservation of phosphorylation sites in this region across species. **d** Growth curve for Rps14a T119A and S123A mutants in 25 °C in SC media. Source data are provided as a Source Data file. **e** In situ hybridization with a Cy3-labeled oligonucleotide complementary to the 5′ sequence portion of ITS1 was assayed in 30 and 20 °C. **f** Structural representation of contacts between the hotspot region of Rps14a (represented in gray) and the ATPase domain of Fap7 (represented in orange)

rRNA processing (Supplementary Fig. 3a). It could be that at 16 and 20 °C the phospho-mutant Rps14a-T119A may not be able to activate Fap7, resulting in impaired incorporation of uS11 and eS26 on the assembling pre-ribosome and consequently fail to undergo 20S pre-RNA processing in the cytoplasm. Despite being growth impaired at 39 °C (Supplementary Fig. 2a), the S123A mutant showed neither a cytoplasmic Cy3-ITS1 signal nor increased levels of 20S pre-RNA (Supplementary Fig. 3).

## Discussion

In this study, we have identified regions within domain families that are recurrently phosphorylated across different instances of a protein domain family in different proteins and species. Given that often there are multiple copies of the same domain within each genome, a domain-centric conservation analysis has increased statistical power over studying conservation across orthologs. However, domain-centric approaches will tend to identify only features that are conserved across members of the domain family and will tend to miss gene-specific features. This limitation is well illustrated with the protein kinase family. While

75% of human phosphosites with known regulatory roles in kinases are found in the activation loop, several other regulatory sites are found across most of the kinase domain sequence (Fig. 2). While conservation provides a signal of functional importance, this approach will miss functionally important phosphosites that are not strongly conserved. In addition, the domain models only cover a fraction of the proteome and most phosphorylation occurs outside these regions. Therefore domain hotspots are useful to identify functional important phosphosites but will tend to miss gene specific regulation and cannot provide information for phosphosites outside domain regions.

Although phosphorylation hotspots cover only a fraction of phosphosites, the annotation of these regions, in the context of protein structures, allows us to study how protein domain functions can be regulated. We observed that these hotspots are enriched in positions that are at interfaces or near catalytic residues. This suggests that controlling interactions and regulating access to the catalytic centre may be common mechanisms by which phosphorylation can tune the function of protein domains. For some enzymes that catalyze phospho-transfer reactions via phospho-enzyme intermediates (i.e., ALP, NDK, PGM/PMM) we

observed re-current detection of phosphorylation of the catalytic residue or neighboring amino-acids. These cases may often represent phosphorylation that is not catalyzed by protein kinases but instead an intermediate enzymatic step or autophosphorylation. This suggests also that mass spectrometry based approaches can be used to track such enzyme reactions via their intermediate phosphorylation states.

With our analysis we were able to identify in total 241 domain phosphorylation hotspots. In this study, we analyzed phenotypes associated with two point mutants within the C-terminal tail of uS11 (yeast Rps14). We found that the T119A mutant exhibits reduced growth rate at 20 and 16 °C and accumulates 20S pre-rRNA in the cytoplasm. This conserved residue is located at the base of uS11 C-terminal tail. Mutants in the C-terminal tail have been previously shown to also accumulate 20S in the cytoplasm[40]. The C-terminal region of uS11 triggers activation of Fap7-ATPase, whose essential function is to pre-fabricate an uS11:eS26 complex, and co-deposit the ribosomal proteins onto the assembling pre-ribosome in the nucleolus[38]. The ribosomal protein eS26 clamps the 3′ end of rRNA at the site where the endonuclease Nob1 cleaves the 20S pre-rRNA into a mature 18S rRNA[41]. eS26-depletion impairs 20S pre-rRNA processing in the cytoplasm[41] suggesting that the clamping of the 3′ end of rRNA is critical for the endonucleolytic cleavage. Thus, failure to deposit eS26:uS11 on the pre-ribosome does not affect nuclear export of these particles to the cytoplasm. Moreover, these aberrant 40S pre-ribosomes engage with mature 60S subunits to form 80S-like particles, but are unable to process 20S pre-rRNA and therefore accumulate in the cytoplasm[36,40]. Thus, it is tempting to speculate that the underlying 20S pre-RNA processing impairment could arise due to the inability of the T119A mutant to efficiently activate Fap7-ATPase, a critical event to deposit uS11:eS26 onto the assembling pre-ribosome. Notably, even though Rps14a-S123A phosphomutant shows slow growth phenotype at 39 °C and is sensitive to CHX, we did not observe any impairment in nuclear export of 40S pre-ribosomes or cytoplasmic 20S pre-RNA processing. However, given the CHX sensitivity, it could be that this region within uS11 plays a critical role during translation.

The identification of phosphorylation hotspots across a diverse set of domains suggests a widespread ancient role for control of protein domains by phosphorylation in eukaryotic species. It remains to be studied how these regulatory regions arise during evolution. These hotspot regions and annotations are provided in Supplementary Data 1 as a resource for future studies. Jointly analyzing phosphoproteomics and structural data has allowed us to study the potential functions for the phosphorylation of different regions. However, a structural characterization of the role of such phosphosites will require experimentally determining structures in the phosphorylated form. These studies are typically difficult to perform since it is not straightforward to obtain large amounts of purified phospho-protein, in particular in a residue-specific manner. However, recent progress in genetically encoded phosphorylated residues in protein expression systems[42,43] should make these studies more feasible. Such studies can in-turn spur the rational design of novel phosphorylation switches.

## Methods

**Phosphorylation data sources and compilation.** Phosphorylated residues *Homo sapiens*, *Mus musculus*, and *Rattus norvegicus* were obtained from the Phospho-SitePlus database[5]. Phosphorylation data for 6 Drosophila species (*Drosophila ananassae*, *Drosophila melanogaster*, *Drosophila pseudoobscura*, *Drosophila simulans*, *Drosophila virilis*, and *Drosophila yakuba*) was obtained from the iProteinDB in FlyDB database[44]. Two additional metazoan phosphoproteomes were obtained from published studies for *Caenorhabditis elegans*[45] and *Trichoplax adhaerens*[46]. Phosphosites for 18 fungal species (*S. cerevisiae*, *Saccharomyces paradoxus*, *Saccharomyces mikatae*, *Saccharomyces kudriavzevii*, *Saccharomyces bayanus*, *Naumovozyma castellii*, *Candida glabrata*, *Vanderwaltozyma polyspora*,

*Zygosaccharomyces rouxii*, *Kluyveromyces lactis*, *Lachancea kluyveri*, *Lachancea waltii*, *Lachancea thermotolerans*, *Komagataella(Pichia) pastoris*, *Meyerozyma guilliermondii*, *Candida albicans*, *Scheffersomyces stipitis*, *Schizosaccharomyces pombe*) were obtained from Studer and colleagues[9]. An additional fungal phosphoproteome was added for *Magnaporthe oryzae*[47]. Plant phosphosites for 4 species (*Arabidopsis thaliana*, *Glycine max*, *Medicago truncatula*, and *Oryza sativa*) was retrieved from the P3DB database[48] and additional plant species were retrieve from selected publications for *Brachypodium distachyon*[49], *Chlamydomonas reinhardtii*[50], and *Selaginella moellendorffii*[51]. Information for 3 Apicomplexa species (*Plasmodium falciparum*, *Plasmodium berghei*, *Toxoplasma gondii*) were compiled from phosphoproteomic studies for these species[52,53]. For all species we removed potential redundancies to avoid assigning the same phosphosites to multiple sequences which could cause false enrichments in proteins rich in isoforms. For *H. sapiens*, *M. musculus*, and *R. norvegicus* data was retrieved from PhosphoSitePlus, we removed isoform redundancy by using only the canonical sets of proteins as defined by UniProt. For other species we filtered out redundant peptides by removing identical 11 amino-acid peptides centered on reported phosphosite positions. The total list of phosphosites compiled for this study is shown in Fig. 1.

**Domain mapping, alignment, and hotspot predictions.** For all of the analyzed protein sequences we used PfamScan to predict Pfam domains. The PfamScan option for predicting catalytic residues was used to retrieve annotations on these types of residues. For each Pfam domain, all corresponding sequences having at least 1 phosphorylation site mapped to them were selected and aligned using MAFFT (version 7, using the G-INS-i option)[54]. In order to identify alignment regions containing more phosphosites than expected by chance we used a permutation strategy to generate a null background. We first count the observed phosphosites for a given region of a Pfam domain using a rolling window with a fixed size of 5 positions. We chose to use a window instead of individual positions due to the uncertainty in phosphosite localization within the phosphopeptides; evolutionary drift whereby the phosphorylation of nearby residues could have the same outcome; potential uncertainty in the alignment. To generate a null background model we randomly select phospho-acceptor residues (serine, threonine, and tyrosine) within the alignment respecting the total number of acceptor residues for the 3 amino-acid types. Permutations were repeated 100 times and for each position in the alignment and an expected median and standard deviation of phosphorylation were calculated. The observed values were converted to z-scores using the permutation information and then to p-values using the survival function of the normal distribution from scipy (scipy.stats.norm.sf)[55]. Only enrichment over random was considered, not depletion and the Bonferroni correction was used to account for multiple testing globally. In addition, to avoid the identification of positions with a low effect size a cut-off of an average of 2 phosphosites per position was used. Finally, contiguous positions were merged to identify domain regions of interest with added ±2 positions on either side that were defined as phosphorylation hotspot regions.

**Identification of representative structures.** For each Pfam domain having a significant hotspot region we obtained 3D structures available in the PDB. We then identified and selected the corresponding Pfam domain regions within each structure excluding the remainder. The structures were filtered to exclude those with gaps larger than 1 amino-acid and those shorter than 70% of the longest structural model for a given domain family. Each set of structures representing a domain were structurally clustered with MaxCluster with single linkage (http://www.sbg.bio.ic.ac.uk/~maxcluster) and for each domain one structure was selected from the cluster representing the most common conformation. In order to map the hotspot information to structures, the sequence of the representative structure of each domain was aligned to the corresponding Pfam domain sequences using MAFFT. In some instances the structural model did not cover a predicted hotspot region.

**Annotation of interface, catalytic, regulatory, accessibly, and disordered residues.** For the selected structural models we obtained water accessibility using naccess, disordered prediction with DISOPRED[27] and catalytic residues from Pfam (PfamScan). For interface contact information we used data from 3DID[56]. For a given Pfam domain in 3DID we calculated, for each residue, the number of times this residue is found in contact of another protein chain (intra-chain contacts were ignored). We considered a position within a Pfam domain family to be at an interface if it was found forming contacts with other proteins in 10 or more structures. This cut-off was used to obtain a high confidence list of Pfam domain interface positions but similar results were observed considering more lenient definitions of interface positions. A set of human phosphorylation sites known to have regulatory roles were obtained from PhosphoSitePlus[57].

**S. cerevisiae phospho-deficient strain construction.** Phospho-deficient mutants Rps14a T119A and S123A were constructed using the Y8205 background strain (*MATα, his3Δ1; leu2Δ0; ura3Δ0; MET15+; LYS2+; can1Δ::STE2pr- SpHIS5; lyp1Δ:: STE3pr-LEU2*, from Krogan Lab - UCSF). SceI endonuclease (from pND32 plasmid, from Knop lab—ZMBH, Heidelberg) was integrated at the mutated leu2Δ0 locus (Y8205 + leu2Δ0:: natNT2-Gal1pr-I-SceI). The point mutations T119 or S123

were introduced into the *RPS14A* endogenous locus and the URA3 marker after the stop codon. The URA3 marker was flanked by SceI recognition sites to enable its removal by the Galactose inducible SceI endonuclease[58]. Point mutations were verified by sequencing.

**Serial spot dilution assay in *S. cerevisiae*.** Yeast strains were grown on agar plates and individual colonies of each strain were picked and arrayed in 96-well plates containing the synthetic SC medium and incubated overnight. The strains were then serially diluted four times at one in 20 dilutions in 96-well plates filled with 160 μl sterile ddH$_2$0, the dilutions were performed using a Bio mek FXp liquid handler. The diluted cells were then immediately spotted onto SC + condition agar plates using a V&P scientific (VP 405) 96 format manual pinning tool. The agar plates were incubated for 48 and 72 h and imaged.

**Fluorescence in situ hybridization and microscopy.** Cells were grown to mid-logarithmic phase and fixed with 4% formaldehyde in 0.1 M potassium phosphate buffer. Cells were then converted to spheroplasts using 0.1 M potassium phosphate buffer containing 1.2 M sorbitol and 500 μg of zymolase. Spheroplasts were washed in 2× SSC buffer and incubated overnight at 37 °C with Cy3-labeled oligonucleotide probe (5′-Cy3-ATG CTC TTG CCA AAA CAA AAA AAT CCA TTT TCA AAA TTA TTA AAT TTC TT-3′) that is complementary to the 5′ portion of ITS1[59]. DNA was stained with 0.5 μg/ml DAPI. Early biogenesis defect of small ribosomal subunit was determined by localization of uS5-GFP expressed from centromeric plasmid pRS316-*RPS2-GFP*[59,60]. Cells were visualized using a DMI6000 microscope (Leica, Germany) equipped with a HCX PL Fluotar 63×/1.25 NA oil immersion objective (Leica, Germany). Images were acquired with a fitted digital camera (ORCA-ER; Hamamatsu Photonics, Japan) and Openlab software (Perkin-Elmer, USA).

**RNA extraction from yeast and northern blot**. RNA was extracted with Trizol reagent following a standard protocol. Three micrograms of total RNA were separated on a 1.2% agarose/formaldehyde gel for 1.5 h at 200 V. For northern blot analysis, rRNA were blotted onto a Hybond-XL (Amersham, UK) membrane by capillary transfer and probed for 20S (5′-GGTTTTAATTGTCCTATAA-CAAAAGC) using radioactively labeled probes (synthesized by Microsynth). rRNA was detected using phosphoimaging screens (GE Healthcare).

**Reporting summary**. Further information on research design is available in the Nature Research Reporting Summary linked to this article.

## Data availability

The compiled phosphorylation sites, domain alignments used for the calculation of the phosphorylation hotspots are available in GitHub at https://github.com/evocellnet/ptm_hotspots. The source data underlying Fig. 6d, Supplementary Figs. 2b and 3c are provided as a Source Data file. A reporting summary for this Article is available as a Supplementary Information file. All other data supporting the findings of this study are available from the corresponding author on reasonable request.

## Code availability

A Python script for the calculation of the phosphorylation hotspots is available in GitHub at https://github.com/evocellnet/ptm_hotspots.

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

## Acknowledgements

R. Sopko was a Special Fellow of the Leukemia and Lymphoma Society. N.P. is a Howard Hughes Medical Institute investigator. The authors would like to thank Center for Microscopy and Image Analysis, University of Zurich, for access to fluorescence microscope, all members of Panse Laboratory, Institute of Medical Microbiology and UZH for continued support. V.G. Panse is supported by grants from the Swiss National Science Foundation, NCCR RNA & Disease, Novartis Foundation, Olga Mayenfisch Stiftung, and a Starting Grant Award from the European Research Council (EUR-IBIO260676). M.O. is supported by Boehringer Ingelheim Fonds PhD fellowship. P.B. and M.J.S. are supported by a Starting Grant Award from the European Research Council (ERC-2014-STG 638884 PhosFunc). The authors thank Alex Bateman and members of the Beltrao group for helpful discussions and Michael Knop for the pND32 plasmid.

## Author contributions

M.J.S. performed computational analyses. C.V., M.S. and B.P.B. created the *S. cerevisiae* mutant strains and performed growth experiments. M.O. experimentally characterized the *RPS14A* mutant. D.O. and R.A.S. provided assistance with computational analyses and compiled phosphoproteomic data. R.S. and N.P. provided access to phosphorylation data. V.G.P. and P.B. designed experiments and computational analyses. M.J.S. and P.B. conceived the project. M.J.S. and P.B. wrote the paper with the assistance of all authors.

## Additional information

**Competing interests:** The authors declare no competing interests.

