## [Peer Review File · Nature Communications]

Reviewers' comments:

Reviewer #1 (Remarks to the Author):

General comments

The authors examine the conservation of several thousands of phosphorylation sites across 40 eukaryotic species. Evolutionarily conserved sites are known to be functionally important and with a systematic structural analysis, the authors show this specifically in the context of phosphorylation sites. The authors find that the conserved phosphorylation sites called 'hotspots' are disproportionately located on the interaction surfaces of proteins and near catalytic centers. Collectively, the findings underscore the earlier known aspect of protein function but overall does not provide significant advance beyond this aspect. The scope of the dataset and the scale of the analyses are impressive and provide a significant resource. Unfortunately, the major findings only confirm previous observations made on a smaller set.

Major comments

1. Extent of novelty

At the core of the study, the authors argue that the phosphosites that are located at the conserved regions ("hotspots") tend to be functionally important (Page 1 Line 28). This argument appears to be a special case of the general notion that conserved sites and domains tend to be functionally important. The idea of using conservation scores as a predictor of the functionally important sites, in general, is already explored in previous studies [1-3]. Although the study provides systematic analysis of phosphorylation sites and the data is the most extensive to date, it is difficult to see significant advance of the current study over previous ones.

2. Methodology and data

On page 14 line 33-34, the authors posit that their data would be a resource for future studies. This is certainly the case. However, the code itself may also be helpful for future studies. It would allow the readers to test it on their data, update it to accommodate to new data and possibly adapt author's method in different contexts. Because the methodology is at the core of the extent of novelty in the study, I would highly recommend deposition of the code along with the manuscript.

The initial compilation of the phosphosites is central to the conclusion drawn from the author's analysis and given the possibilities of sampling biases in such studies, it would be useful to see the

actual compilation of phosphosites along with the manuscript. Figure 1 shows the species from where the phosphosites were compiled and the proportion of them found within Pfam domains. However, the text (Page 15, Line 25) indicates that they are the list of phosphosites compiled.

Minor comments

The use of commas/semicolons would improve readability of following statements.

1. Page 3 Line 8
2. Page 5 Line 2-5
3. Page 15 Line 39-41

References

1. Panchenko AR, Kondrashov F, Bryant S. Prediction of functional sites by analysis of sequence and structure conservation. *Protein Science*. 2004 Apr;13(4):884-92.
2. Capra JA, Singh M. Predicting functionally important residues from sequence conservation. *Bioinformatics*. 2007 May 22;23(15):1875-82.
3. Capra JA, Laskowski RA, Thornton JM, Singh M, Funkhouser TA. Predicting protein ligand binding sites by combining evolutionary sequence conservation and 3D structure. *PLoS computational biology*. 2009 Dec 4;5(12):e1000585.

Reviewer #2 (Remarks to the Author):

Reviewer's comments, manuscript Nat Communications, Strumillo et al

In this manuscript, Strumillo et al combine phosphosites from 40 eukaryotic species to identify conserved functional phosphorylation "hotspot" regions within domain families and further mapping these regions onto structural data. The authors observed that hotspot regions were often located at interfaces or near catalytic residues. To support their analysis with an experimentally validated example, the analyzed the functional importance of two phospho-sites found within a C-terminal hotspot in the yeast ribosomal protein RpS14/uS11.

While the paper is very interesting, well described and detailed, the functional analysis of the Rps14A hotspot – which is really a proof-of-principle experiment in which the accuracy of their predictions is validated – is not convincing as to the assays used and conclusions drawn from them. For publication in Nature Communications, additional experiments are necessary to show that the two phospho-sites really affect the particular step in ribosome biogenesis that the authors claim. For explanations, see my comments below.

Major points:

On page 12, the authors describe the analysis of two phospho-site mutations in yeast RpS14A, S123A and T119A, which are both located within a determined hotspot, to determine a potential importance for either site in translating ribosomes and/or ribosome biogenesis.

Both mutants show a sensitivity to cycloheximide, however, the authors only assay one mutant further with regards to ribosome biogenesis. There are a number of issues with the used assays that do not allow them to fully support their conclusions.

1) The u55-GFP reporter assay mostly points towards ribosome export defects, not ribosome assembly/processing ones (although sometimes they can be the cause for non-exported ribosomes). Ribosome maturation has many parallel pathways and effects on biogenesis cannot simply be discerned by such an assay. I suggest the authors carry out a Northern blot analysis of the pre-ribosomal RNA to see if the two mutations in RpS14 cause an effect on ribosome biogenesis (the precursors not just the mature rRNAs, which is very stable). Moreover, DAPI or a nucleolar marker should have been used since it is very hard to make out what is nucleus and what vacuole in the images presented in Sup Fig2.

2) I am also a bit wary about the ‘cold-sensitive’ designation used here. Cold-sensitivity in yeast is usually assayed at 16C, not 20C. 20-25C is a very common temperature for *S.cerevisiae* (outside the lab). Did the authors test lower temperatures? Higher ones (i.e.37C)? The cells in the u55 assay do not look at that different at 20C from WT in SupFig2.(especially given the lack of DAPI staining) – why do the authors conclude it the T119A mutant exhibits a cold sensitivity? Only based on the growth curve? Was the doubling time significantly different between the two mutants?

3) Especially since the authors then point out that cytoplasmic 20S processing may be affected (late rather than early 40S maturation), a Northern blot analysis is definitely required. The FISH experiment is not sufficient to state clearly that this is what happens as the diffused cytoplasmic ITS1 signal in the mutant at 20C suggests that unprocessed 20S pre-rRNA may be incorporated into mature 40S subunits. This may also explain the growth defect under cycloheximide that the authors observe.

4) Even though the growth curve was not as striking than T119A, the CHX effect was also observed with S123A (even more than with T119A) – why was this mutant not tested for biogenesis defects?

5) The authors state that “Interestingly, RPS14A has a paralog - RPS14B that was not deleted or mutated for these studies, meaning that rps14a T119A mutant might act in a dominant negative manner.” Was the P-site hotspot also found in the paralog?

6) The authors state: “This tail region was shown to make contacts with the ATPase domain of Fap7 (Figure 6E) and the C-terminal region of uS11 was demonstrated to activate the ATPase Fap7, a critical step to release and deposit uS11 and its interacting partner eS26 into its rRNA binding site (Peña et al, 2016). It seems likely that the phospho-mutant rps14a T119A may not be able to activate Fap7”. How would that affect processing of the 20S at site D by Nob1 then? And translation? This should be discussed. It is also notable that while the authors mention their proof-of-principle experiment on Rps14 in their abstract it is not mentioned in their Discussion section.

Minor Points:

- Page 12: The following sentence needs to be rephrased for clarity: “Based on the initial growth defect we tested but saw no phenotype in the early steps of ribosome assembly using a uS5-GFP reporter assay (Supplementary Figure 2).”
- The authors should briefly explain what 6-azauracil (6AU) and cycloheximide (CHX) do, for the more general reader.

Reviewer #3 (Remarks to the Author):

The study is interesting. However it needs to be expanded:

1. The authors should show another array of 'control experiments' which specifically deals with study/literature bias:

a. Sites biologists have chosen to study functionally tend to be most conserved.

b. Sites we can study functionally tend to depend on antibodies which in turn tend to be raised against conserved sites or 'accessible sites'

c. Studies and study techniques tend to focus on most abundant proteins. This means there is a bias towards sites in more abundant proteins.

etc.

These biases are STUDY BIASES not BIOLOGICAL/EVOLUTIONARY BIASES. Thus the authors must include analysis of how this affect their results.

In particular it is important to analyse how this affect conclusions for sites that are less conserved, which may very well still have function.

Also the authors should consider that some sites may have functions most relevant for certain 'realms' of evolution, this is well known for tyrosine in metazoans; but there could be other areas.

2. We know that biological systems require operational freedom and the ability to 'evolve' can depend on having 'options' thus just because there is no function known or visible a site can be important for an evolutionary trajectory or enable another site to obtain a function.

All these considerations should be discussed and considered. Some of this can be done by using number of PUBMED ID's as a normalisation factor or other literature bias measurements to normalize/compare etc.

Point by point response to the reviewers' comments with our responses highlighted in blue

Reviewers' comments:

Reviewer #1 (Remarks to the Author):

General comments

The authors examine the conservation of several thousands of phosphorylation sites across 40 eukaryotic species. Evolutionarily conserved sites are known to be functionally important and with a systematic structural analysis, the authors show this specifically in the context of phosphorylation sites. The authors find that the conserved phosphorylation sites called 'hotspots' are disproportionately located on the interaction surfaces of proteins and near catalytic centers. Collectively, the findings underscore the earlier known aspect of protein function but overall does not provide significant advance beyond this aspect. The scope of the dataset and the scale of the analyses are impressive and provide a significant resource. Unfortunately, the major findings only confirm previous observations made on a smaller set.

Major comments

1. Extent of novelty

At the core of the study, the authors argue that the phosphosites that are located at the conserved regions ("hotspots") tend to be functionally important (Page 1 Line 28). This argument appears to be a special case of the general notion that conserved sites and domains tend to be functionally important. The idea of using conservation scores as a predictor of the functionally important sites, in general, is already explored in previous studies [1-3]. Although the study provides systematic analysis of phosphorylation sites and the data is the most extensive to date, it is difficult to see significant advance of the current study over previous ones.

We thank the reviewer for noting that this is the most extensive and systematic analysis to date of conserved phosphorylation sites. We certainly agree with the notion that using conservation to identify functional important residues is not new. We would even say that the notion dates back to origin of bioinformatics with the works of Dayhoff in the 1960s. The idea that conservation can be used to identify functionally important residues is not what we are proposing here to be the novelty. The novelty here is the identification of the regulatory regions themselves within the domain families. While some of these regions

have been previously described, such as the activation loop of the protein kinase family, the majority of the regulatory domain regions suggested here are novel. The analysis showing these regions tend to be at interfaces or new catalytic residues is meant to further give credence that these regions are indeed likely to be regulatory and phosphorylation of such regions can control domain functions. Each of such regions is a novel finding in itself and we then followed up in more detail one example in a ribosomal protein.

2. Methodology and data

On page 14 line 33-34, the authors posit that their data would be a resource for future studies. This is certainly the case. However, the code itself may also be helpful for future studies. It would allow the readers to test it on their data, update it to accommodate to new data and possibly adapt author's method in different contexts. Because the methodology is at the core of the extent of novelty in the study, I would highly recommend deposition of the code along with the manuscript.

The initial compilation of the phosphosites is central to the conclusion drawn from the author's analysis and given the possibilities of sampling biases in such studies, it would be useful to see the actual compilation of phosphosites along with the manuscript. Figure 1 shows the species from where the phosphosites were compiled and the proportion of them found within Pfam domains. However, the text (Page 15, Line 25) indicates that they are the list of phosphosites compiled.

We agree with the reviewer and have made the code and data available for re-use by others. The code can be found in a Github repository (https://github.com/evocellnet/ptm_hotspots) along with instructions on how to use. In order to allow others to reuse the code we also made available the PFAM domain alignments and the list of phosphosites. As the author notes, the coverage of protein phosphorylation in each species is uneven due to the mass spectrometry approaches not reaching full coverage and some species having been more often assayed. We note however that such differences in coverage should not affect the accuracy of the approach since we are not attempting to define species or clade specific regulatory regions. We have updated the manuscript to indicate the availability of the phosphosite information, alignments and code.

Minor comments

The use of commas/semicolons would improve readability of following statements.

1. Page 3 Line 8
2. Page 5 Line 2-5
3. Page 15 Line 39-41

We thank the reviewer for the useful suggestions. We have revised the text to improve the readability in these sections.

Reviewer #2 (Remarks to the Author):

Reviewer's comments, manuscript Nat Communications, Strumillo et al

In this manuscript, Strumillo et al combine phosphosites from 40 eukaryotic species to identify conserved functional phosphorylation "hotspot" regions within domain families and further mapping these regions onto structural data. The authors observed that hotspot regions were often located at interfaces or near catalytic residues. To support their analysis with an experimentally validated example, they analyzed the functional importance of two phospho-sites found within a C-terminal hotspot in the yeast ribosomal protein RpS14/uS11.

While the paper is very interesting, well described and detailed, the functional analysis of the Rps14A hotspot – which is really a proof-of-principle experiment in which the accuracy of their predictions is validated – is not convincing as to the assays used and conclusions drawn from them. For publication in Nature Communications, additional experiments are necessary to show that the two phospho-sites really affect the particular step in ribosome biogenesis that the authors claim. For explanations, see my comments below.

We thank the reviewer for the positive comments regarding the work. As requested, we have addressed the specific concerns regarding the Rps14A hotspot with additional experiments and clarifications.

Major points:

On page 12, the authors describe the analysis of two phospho-site mutations in yeast RpS14A, S123A and T119A, which are both located within a determined hotspot, to determine a potential importance for either site in translating ribosomes and/or ribosome biogenesis.

Both mutants show a sensitivity to cycloheximide, however, the authors only assay one mutant further with regards to ribosome biogenesis. There are a number of issues with the used assays that do not allow them to fully support their conclusions.

1) The uS5-GFP reporter assay mostly points towards ribosome export defects, not ribosome assembly/processing ones (although sometimes they can be the cause for non-exported ribosomes). Ribosome maturation has many parallel pathways and effects on biogenesis cannot simply be discerned by such an assay. I suggest the authors carry out a Northern blot analysis of the pre-ribosomal RNA to see if the two mutations in RpS14 cause an effect on ribosome biogenesis (the precursors not just the mature rRNAs, which is very stable). Moreover, DAPI or a nucleolar marker should have been used since it is very hard to make out what is nucleus and what vacuole in the images presented in Sup Fig2.

We would like to clarify the use of uS5-GFP and Cy3-ITS1 reporters in monitoring 40S pre-ribosome export and cytoplasmic 20S pre-rRNA processing, respectively. 40S pre-

ribosomes containing 20S pre-rRNA are assembled in the nucleolus/nucleus, and exported into the cytoplasm where they undergo final maturation before achieving translation competence. One final step includes processing of immature 20S pre-rRNA into mature 18S rRNA by the endonuclease Nob1 within an 80S-like particle formed between a mature 60S subunit and the 40S pre-ribosome (Strunk et al., 2012 Cell; Lebaron et al. 2012 NSMB). Accumulation of 20S pre-rRNA in yeast mutants therefore reflects either (1) impaired nuclear export of 40S pre-ribosomes, or that (2) the exported 40S pre-ribosomes fail to undergo 20S pre-rRNA processing in the cytoplasm. We can distinguish these possibilities using the uS5-GFP reporter that monitors nuclear export of 40S pre-ribosomes, and a FISH assay employing the Cy3-ITS1 probe that hybridizes with ITS1, and monitors 20S pre-rRNA processing in the cytoplasm. If a mutant strain exhibits nuclear location of uS5-GFP, then the observed increased 20S pre-rRNA levels are very likely due to nuclear export impairment of 40S pre-ribosomes. In this case, Cy3-ITS1 probe location will overlap with nucleoplasmic DAPI stain (for e.g. in the *xpo1-1* mutant, *yrb2Δ* cells; Faza et al., 2012 PLoS Genetics). In contrast, if a mutant does not accumulate uS5-GFP in the nucleus, but still shows increased 20S pre-rRNA levels, this reflects a cytoplasmic 20S pre-rRNA processing defect. In this case, the Cy3-ITS1 probe will show a strong cytoplasmic staining (Tsr2-depleted and Fap7-depleted cells in Schütz et al., 2014 eLife; Pena et al., 2016 eLife). The *rps14aT119A* mutant reported in this study (Figure 6E) as well as several C-terminal mutants of Rps14 (R134A, G135A, R137A) previously characterized by the Woolford laboratory (Jakovljevic et al., 2004). The *rps14aT119A* mutant does not accumulate uS5-GFP in the nucleus. FISH experiments using a Cy3-ITS1 probe show a strong cytoplasmic staining indicating impaired processing of 20S pre-rRNA in the cytoplasm. As requested, we have now performed Northern analyses, which show strongly increased 20S pre-rRNA levels (Supplementary Figure 2B). We have now better explained this in the Results Section.

2) I am also a bit wary about the 'cold-sensitive' designation used here. Cold-sensitivity in yeast is usually assayed at 16C, not 20C. 20-25C is a very common temperature for *S.cerevisiae* (outside the lab). Did the authors test lower temperatures? Higher ones (i.e.37C)? The cells in the uS5 assay do not look at that different at 20C from WT in SupFig2.(especially given the lack of DAPI staining) – why do the authors conclude it the T119A mutant exhibits a cold sensitivity? Only based on the growth curve? Was the doubling time significantly different between the two mutants?

Based on dot-spot analyses, and impaired doubling time as monitored by growth curves, we conclude that the *rps14aT119A* is cold sensitive since it is strongly impaired in growth at 16°C and 20°C, but not at 30°C and higher temperatures 37°C and 39°C. We added a new Supplementary figure 2A with the dot-spot sensitivity at lower temperatures. The doubling time is significantly different as can be seen in the growth curve experiments that were done with replicates and error bars represent the technical variability of the assay (Figure 6D). We have now precisely phrased “cold sensitive” in the text to: impaired growth at 16°C and 20°C.

3) Especially since the authors then point out that cytoplasmic 20S processing may be affected (late rather than early 40S maturation), a Northern blot analysis is definitely required. The FISH experiment is not sufficient to state clearly that this is what happens as

the diffused cytoplasmic ITS1 signal in the mutant at 20C suggests that unprocessed 20S pre-rRNA may be incorporated into mature 40S subunits. This may also explain the growth defect under cycloheximide that the authors observe.

Please see point 1 for clarification of this concern. Northern analyses show strongly increased 20S pre-rRNA levels for *rps14aT119A* mutant at 20°C (Supplementary Figure 2B). Based on the cytoplasmic u5-GFP location, increased cytoplasmic signal of Cy3-ITS1 and complementary Northern analyses, we conclude that late 40S maturation i.e. 20S pre-rRNA processing in the cytoplasm is impaired.

4) Even though the growth curve was not as striking than T119A, the CHX effect was also observed with S123A (even more than with T119A) – why was this mutant not tested for biogenesis defects?

We have analysed the *rps14aT119A* mutant for maturation defects due to the striking cold sensitive phenotype, a common phenotype associated with impaired ribosome assembly. We investigated growth *rps14aS123A* mutant at different temperatures, and observed a growth defect only at 39°C by dot-spot analyses (Supplementary Figure 2A). We have performed similar analysis for *rps14aS123A* mutant at 20°C, 30°C and 39°C. The *rps14aS123A* mutant was not impaired in nuclear export of 40S pre-ribosomes, and cytoplasmic 20S pre-rRNA processing as judged by the u5-GFP assay, FISH and Northern blot analyses (Supplementary Figure 3). The reasons for the impaired growth at 39°C remains unclear. Given the sensitivity to CHX, we suspect that *rps14aS123A* mutant might be impaired in specific steps of the translation cycle. It is intriguing that despite the proximity of the two phospho-sites T119 and S123, the two mutants do not share the same phenotypes.

5) The authors state that “Interestingly, RPS14A has a paralog - RPS14B that was not deleted or mutated for these studies, meaning that *rps14a T119A* mutant might act in a dominant negative manner.” Was the P-site hotspot also found in the paralog?

The phosphorylation hotspot is calculated at the protein domain level as this is how we derive the statistical power to identify these regions across many copies of the domains in different species. So, in this regard, the RPS14B also has the hotspot defined in the same region.

6) The authors state: “This tail region was shown to make contacts with the ATPase domain of Fap7 (Figure 6E) and the C-terminal region of uS11 was demonstrated to activate the ATPase Fap7, a critical step to release and deposit uS11 and its interacting partner eS26 into its rRNA binding site (Peña et al, 2016). It seems likely that the phospho-mutant *rps14a T119A* may not be able to activate Fap7”. How would that affect processing of the 20S at site D by Nob1 then? And translation? This should be discussed. It is also notable that while the authors mention their proof-of-principle experiment on RpS14 in their abstract it is not mentioned in their Discussion section.

The ribosomal protein eS26 clamps the 3' end of rRNA at the site where the endonuclease Nob1 processes 20S pre-rRNA into 18S rRNA (Schütz et al., 2014 eLife). eS26-depletion

impairs 20S pre-rRNA processing in the cytoplasm (Schütz et al., 2014) suggesting that the clamping of the 3' end of rRNA is critical for the endonucleolytic cleavage.

eS26 is delivered to the assembling pre-ribosome in complex with its neighboring ribosomal protein uS11. The essential function of the ATPase Fap7 is to pre-fabricate an uS11:eS26 complex, and co-deposit the ribosomal proteins onto the assembling pre-ribosome in the nucleolus (Pena et al., 2016). This deposition step involves activation of Fap7-ATPase, which is triggered by the C-terminal region of uS11 (Helmich et al., 2013). Failure to deposit eS26:uS11 on the pre-ribosome does not affect nuclear export of these particles to the cytoplasm (Pena et al., 2016). In the cytoplasm, these aberrant particles engage with mature 60S subunits to form 80S-like particles, they are unable to process 20S pre-rRNA and accumulate in the cytoplasm (Jakovljevic et al., 2004; Strunk et al., 2012).

We have included this in the Discussion

Minor Points:

- Page 12: The following sentence needs to be rephrased for clarity: "Based on the initial growth defect we tested but saw no phenotype in the early steps of ribosome assembly using a uS5-GFP reporter assay (Supplementary Figure 2)."

We rephrased the sentence in the main text on page 12.

- The authors should briefly explain what 6-azauracil (6AU) and cycloheximide (CHX) do, for the more general reader.

We briefly explained the mechanism of action of 6AU and CHX drugs in the main text (page 12).

Reviewer #3 (Remarks to the Author):

The study is interesting. However it needs to be expanded:

1. The authors should show another array of 'control experiments' which specifically deals with study/literature bias:

a. Sites biologists have chosen to study functionally tend to be most conserved.

b. Sites we can study functionally tend to depend on antibodies which in turn tend to be raised against conserved sites or 'accessible sites'

c. Studies and study techniques tend to focus on most abundant proteins. This means there is a bias towards sites in more abundant proteins.

etc.

These biases are STUDY BIASES not BIOLOGICAL/EVOLUTIONARY BIASES. Thus the authors must include analysis of how this affect their results.

We agree with the reviewer that the set of phosphosites that currently are known to have functions have social biases such as the ones indicated above. In fact we think this underscores the need to develop computational approaches that are not biased in these ways. We don't think these biases have any impact on the work presented in this manuscript. The only point in the manuscript where we use a list of phosphosites with known function is in Figure 1C and 1D. These sites are used just to show that the phosphorylation "hotspot" regions that we identified from the conservation analysis are enriched in phosphosites of previously known function. We don't use the phosphosites of known function in any way to train or define which regions we consider to be hotspots. Independently from such biased phosphosites we also find that the regions of conserved phosphorylation are enriched in functionally important domain positions (i.e. near catalytic and interface residues). Within each domain, the residues that are defined as interface or catalytic are much less likely to be biased by social aspects linked to their discovery. We think these analysis together confirm that the hotspots are enriched in functionally important regions.

In particular it is important to analyse how this affect conclusions for sites that are less conserved, which may very well still have function.

As we discussed above, we don't use the list of known phosphosites to define the regulatory hotspots it is merely used to show that the hotspots are enriched in phosphosites with previously described functions. It is also clearly in line with expectation that highly conserved residues within proteins are more likely to be functionally important when compared to less conserved regions. We don't mean to say that phosphosites outside the hotspot regions do not have a function, we simply claim that those within hotspots are more likely to be important based simply on the evolutionary constraint observed and based on different independent lines of evidence: enrichment of phosphosites of known function, near catalytic and at interface positions. We have further discussed this point raised by the reviewer and the ideas that phosphosites that are not conserved can still have important functional roles.

Also the authors should consider that some sites may have functions most relevant for certain 'realms' of evolution, this is well known for tyrosine in metazoans; but there could be other areas.

This is an important point that is not straightforward to address. We agree with the reviewer that not all hotspot regions are likely to be universal in the tree of life. There will certainly be protein domain families that have acquired phospho-regulatory regions within specific clades. However, in order to search for such events we would sacrifice the statistical power by analyzing subsets of species. Additional method development will be needed in order to make use of the phylogenetic tree and the differences in coverage in the phosphoproteomics of the species to be able to address this question. This would be a very significant expansion of the current work. We discuss in the manuscript this possible expansion of the work.

2. We know that biological systems require operational freedom and the ability to 'evolve' can depend on having 'options' thus just because there is no function known or visible a site can be important for an evolutionary trajectory or enable another site to obtain a function.

This is an important point for discussion. Individual phosphosites that have no function in extant species may serve as future material for selection during changes in selection pressures. However, the focus of the manuscript is not on the individual phosphosites but the regions within the protein domain families that are important for regulation such as the activation loop of kinases.

All these considerations should be discussed and considered. Some of this can be done by using number of PUBMED ID's as a normalisation factor or other literature bias measurements to normalize/compare etc.

As we have stated above, we don't use the phosphosites with known functions to define the hotspot regions in any way. There is no training of machine learning predictors. We simply used them to show that the conserved phosphorylation regions are enriched in phosphosites of known function. In order to address the specific point raised here we binned the phosphosites of known function according to the number of pubmed IDs supporting them. We then repeated the enrichment test with phosphosites of known function having different level of support in the literature. Below we show the ROC curve analysis when we considered all known regulatory phosphosites or when we separate between known regulatory sites supported by 1 or more PMIDS. The results indicate that regions containing highly conserved phosphorylation sites are more likely to be enriched by regulatory regions supported by multiple publications but still strongly enriched in regulatory sites supported by a single publication. In line with the reviewer's point, this could suggest that scientists may be more likely to investigate phosphosites when they are in conserved positions. However, as we have stated, this social bias in no way affects how we determine the phosphorylation hotspots that are based on a statistical analysis of large scale phosphoproteomics.

REVIEWERS' COMMENTS:

Reviewer #1 (Remarks to the Author):

The authors have made the required changes.

Reviewer #2 (Remarks to the Author):

Reviewer's comments, revised manuscript Nat Communications, Strumillo et al

In their revised manuscript, Strumillo et al. have addressed all my concerns and I recommend the manuscript for publication in Nature Communications.

I would ask the authors to correct the minor point below since the text as written now is confusing (i.e., cytoplasmic 20S processing is unusual and a phenotype of the mutant).

Minor Point:

Abstract: the in red added text should read "impaired growth and defective cytoplasmic 20S..."

Reviewer #3 (Remarks to the Author):

The authors have answered my questions/concerns.